# A noise-corrected Langevin algorithm and sampling by half-denoising

## Abstract

The Langevin algorithm is a classic method for sampling from a given pdf in a real space. In its basic version, it only requires knowledge of the gradient of the log-density, also called the score function. However, in deep learning, it is often easier to learn the so-called "noisy score function", i.e. the gradient of the log-density of noisy data, more precisely when Gaussian noise is added to the data. Such an estimate is biased and complicates the use of the Langevin method. Here, we propose a noise-corrected version of the Langevin algorithm, where the bias due to noisy data is removed, at least regarding first-order terms. Unlike diffusion models, our algorithm needs to know the noisy score function for one single noise level only. We further propose a simple special case which has an interesting intuitive interpretation of iteratively adding noise the data and then attempting to remove half of that noise.

## 1  Introduction

A typical approach to generative AI with data in $\mathbb{R}^k$ consists of estimating an energy-based (or score-based) model of the data and then applying a sampling method, typically MCMC. Among the different methods for learning the model of the data, score matching is one of the simplest and has achieved great success recently. In particular, denoising score matching (DSM) is widely used since it is computationally very compatible with neural networks, resulting in a simple variant of a denoising autoencoder (Vincent, 2011). The downside is that it does not estimate the model for the original data but for noisy data, which has Gaussian noise added to it. This is why it is difficult to use standard MCMC methods for sampling in such a case. Henceforth we call the score function for the noisy data the "noisy score function". We emphasize that the noisy score function does not refer to an imperfectly estimated score function which would be noisy due to, for example, a finite sample; it is the exact score function for the noisy data, since this is what DSM gives in the limit of infinite data.

A possible solution to this problem of bias is given by diffusion models (Croitoru et al., 2023; Yang et al., 2023), which are used in many of the state-of-the-art methods (Rombach et al., 2022). Diffusion models need precisely the score function of the noisy data; the downside is that they need it for many different noise levels. Another solution is to estimate the noisy score for different noise levels and then extrapolate (anneal) to zero noise (Song and Ermon, 2019). However, such learning necessitating many noise levels is computationally costly.

One perhaps surprising utility of using the noisy score function for sampling might be that it is a kind of regularized version of the true score function. In particular, any areas in the space with zero data will have a non-zero pdf when noise is added. This might lead to better mixing of any MCMC (Saremi and Hyvärinen, 2019; Song and Ermon, 2019), and even improve the estimation of the score function (Kingma and Cun, 2010).

Here, we develop an MCMC algorithm which uses the noisy score function, and needs it for a single noise level only. This should have computational advantages compared to diffusion models, as well as leading to a simplified theory where the estimation and sampling parts are clearly distinct. In particular, we develop a *noise-corrected Langevin* algorithm, which works using the noisy score function, while providing samples

from the original, noise-free distribution. Such a method can be seamlessly combined with denoising score matching. A special case of the algorithm has a simple intuitive intepretation as an iteration where Gaussian noise is first added to the data, and then, based on the well-known theory by Tweedie and Miyasawa, it is attempted to remove that noise by using the score function. However, only half of the denoising step is taken, leading to an algorithm based on "half-denoising".

## 2 Background

We first develop the idea of denoising score matching based on the Tweedie-Miyasawa theory, and discuss its application in the case of the Langevin algorithm.

### 2.1 Score functions and Tweedie-Miyasawa theory

We start with the well-known theory for denoising going back to Miyasawa (1961); Robbins et al. (1956); see Raphan and Simoncelli (2011); Efron (2011) for a modern treatment and generalizations. They considered the following denoising problem. Assume the observed data $\tilde{\mathbf{x}}$ is a sum of original data $\mathbf{x}$ and some noise $\mathbf{n}$:

$$\tilde{\mathbf{x}} = \mathbf{x} + \mathbf{n} \tag{1}$$

Now, the goal is to recover $\mathbf{x}$ from an observation of $\tilde{\mathbf{x}}$. The noise $\mathbf{n}$ is assumed Gaussian with covariance $\mathbf{\Sigma_n}$.

An important role here is played by the gradient of the log-derivative of the pdf of $\tilde{\mathbf{x}}$. We denote for any random vector $\mathbf{y} \in \mathbb{R}^k$

$$\mathbf{\Psi_y}(\mathbf{y}) = \nabla_{\mathbf{y}} \log p_{\mathbf{y}}(\mathbf{y}) \tag{2}$$

which is here called the *score function*.

One central previous result is the following theorem, often called an "empirical Bayes" theorem. Robbins et al. (1956); Efron (2011) attribute it to personal communication from Maurice Tweedie, which is why it is often called "Tweedie's formula", while it was also independently published by Miyasawa (1961).

**Theorem 1 (Tweedie-Miyasawa)**

$$\mathrm{E}\{\mathbf{x}|\tilde{\mathbf{x}}\} = \tilde{\mathbf{x}} + \mathbf{\Sigma_n}\mathbf{\Psi_{\tilde{x}}}(\tilde{\mathbf{x}}) \tag{3}$$

For the sake of completeness, the proof is given in the Appendix. The Theorem provides an interesting solution to the denoising problem, or recovering the original $\mathbf{x}$, since it gives the conditional expectation which is known to minimize mean-squared error. In practice, the price to pay is that we need to estimate the score function $\mathbf{\Psi_{\tilde{x}}}$.

### 2.2 (Denoising) score matching

Another background theory we need is (non-parametric) estimation of the score function. Consider the problem of estimating the score function $\mathbf{\Psi_x}$ of some data set. Suppose in particular that we use a neural network to estimate $\mathbf{\Psi_x}$. Any kind of maximum likelihood estimation is computationally very difficult, since it would necessitate the computation of the normalization constant (partition function).

Hyvärinen (2005) showed that the score function can be estimated by minimizing the expected squared distance between the model score function $\mathbf{\Psi}(\mathbf{x}; \boldsymbol{\theta})$ and the empirical data score function $\mathbf{\Psi_x}$. He further showed, using integration by parts, that such a distance can be brought to a relatively easily computable form. Such score matching completely avoids the computation of the normalization constant, while provably providing a estimator that is consistent. However, due to the existence of higher-order derivatives in the objective, using the original score matching objective by Hyvärinen (2005) is still rather difficult in the particular case of deep neural networks, where estimating any derivatives of order higher than one can be computationally demanding (Martens et al., 2012). Therefore, various improvements have been proposed, as reviewed by Song and Kingma (2021).

Denoising score matching (DSM) by Vincent (2011) provides a particularly attractive approach in the context of deep learning. Given original noise-free data $\mathbf{x}$, it learns the score function of a noisy $\tilde{\mathbf{x}}$ with noise *artificially* added as in (1). Interestingly, denoising score matching can be derived from Theorem 1. Since the conditional expectation is the minimizer of the mean squared error, we have the following corollary:

**Corollary 1** *Assume we observe both* $\mathbf{x}$ *and* $\tilde{\mathbf{x}}$*, which follow (1) with Gaussian noise* $\mathbf{n}$*. The score function* $\mathbf{\Psi}_{\tilde{\mathbf{x}}}$ *of the noisy data is the solution of the following minimization problem:*

$$\min_{\mathbf{\Psi}} \mathrm{E}_{\tilde{\mathbf{x}},\mathbf{x}}\{\|\mathbf{x} - (\tilde{\mathbf{x}} + \mathbf{\Sigma}_{\mathbf{n}}\mathbf{\Psi}(\tilde{\mathbf{x}}))\|^2\} \tag{4}$$

We emphasize that unlike in the original Tweedie-Miyasawa theory, it is here assumed we observe the original noise-free $\mathbf{x}$, and add the noise ourselves to create a self-supervised learning problem. Thus, we learn the noisy score function by training what is essentially a denoising autoencoder.

The advantage of DSM with respect to the original score matching is that there are less derivatives; in fact, the problem is turned into a basic least-squares regression. The disadvantage is that we only get the noisy score function $\mathbf{\Psi}_{\tilde{\mathbf{x}}}$. This is a biased estimate, since what we want in most cases is the original $\mathbf{\Psi}_{\mathbf{x}}$.

### 2.3 Langevin algorithm

The Langevin algorithm is perhaps the simplest MCMC method to work relatively well in $\mathbb{R}^k$. The basic scheme is as follows. Starting from a random point $\mathbf{x}_0$, compute the sequence:

$$\mathbf{x}_{t+1} = \mathbf{x}_t + \mu\mathbf{\Psi}_{\mathbf{x}}(\mathbf{x}_t) + \sqrt{2\mu}\,\boldsymbol{\nu}_t \tag{5}$$

where $\mathbf{\Psi}_{\mathbf{x}} = \nabla_{\mathbf{x}} \log p(\mathbf{x})$ is the score function of $\mathbf{x}$, $\mu$ is a step size, and $\boldsymbol{\nu}_t$ is Gaussian noise from $\mathcal{N}(\mathbf{0},\mathbf{I})$. According to well-known theory, for an infinitesimal $\mu$ and in the limit of infinite $t$, $\mathbf{x}_t$ will be a sample from $p$. The assumption of an infinitesimal $\mu$ can be relaxed by applying a "Metropolis-adjustment", but that complicates the algorithm, and requires us to have access to the energy function as well, which may be the reason why such a correction is rarely used in deep learning.

If we were able to learn the score function of the original $\mathbf{x}$, we could do sampling using this Langevin algorithm. But DSM only gives us the score function of the noisy data. This contradiction inspires us to develop a variant of the Langevin algorithm that works with $\mathbf{\Psi}_{\tilde{\mathbf{x}}}$ instead of $\mathbf{\Psi}_{\mathbf{x}}$.

## 3 Noise-corrected Langevin algorithm

Now, we proceed to propose a new "noise-corrected" version of the Langevin algorithm. The idea of noise-correction assumes that we have used DSM to estimate the score function of noisy data. This means our estimate of the score function is strongly biased. While this bias is known, it is not easy to correct by conventional means, and a new algorithm seems necessary.

### 3.1 General algorithm and convergence theorem

We thus propose the following noise-corrected Langevin algorithm. Starting from a random point $\mathbf{x}_0$, compute the sequence:

$$\mathbf{x}_{t+1} = \tilde{\mathbf{x}}_t + \mu\mathbf{\Psi}_{\tilde{\mathbf{x}}}(\tilde{\mathbf{x}}_t) + \sqrt{2\mu - \sigma^2}\,\boldsymbol{\nu}_t \tag{6}$$

with

$$\tilde{\mathbf{x}}_t = \mathbf{x}_t + \sigma\mathbf{n}_t, \qquad \mathbf{n}_t \sim \mathcal{N}(\mathbf{0},\mathbf{I}), \qquad \boldsymbol{\nu}_t \sim \mathcal{N}(\mathbf{0},\mathbf{I}) \tag{7}$$

where $\mu$ is a step size parameter. The function $\mathbf{\Psi}_{\tilde{\mathbf{x}}}$ is the score function of the noisy data $\tilde{\mathbf{x}}$ created as in (1), with noise variance being $\mathbf{\Sigma}_{\mathbf{n}} = \sigma^2\mathbf{I}$. (As in previous sections, we use here $\mathbf{x}$ without time index to denote the original observed data, and $\tilde{\mathbf{x}}$ to denote the observed data with noise added, while the same letters with time indices denote related quantities in the MCMC algorithm.)

The main point is that this algorithm only needs the noisy score function $\mathbf{\Psi}_{\tilde{\mathbf{x}}}$, which could be given, for example, by DSM. The weight $\sqrt{2\mu - \sigma^2}$ of the last term in the iteration (6) is accordingly modified from the original Langevin iteration (5) where it was $\sqrt{2\mu}$. Clearly, we need to assume the condition

$$\mu \geq \sigma^2/2 \tag{8}$$

to make sure this weight is well-defined. Another important modification is that the right-hand side of (6) is applied on a noisy version of $\mathbf{x}_t$, adding noise $\mathbf{n}_t$ to the $\mathbf{x}_t$ itself at every step.

Our main theoretical result is the following Theorem, proven in the Appendix:

**Theorem 2** *The original noise-free distribution $p_{\mathbf{x}}$ is the stationary distribution of the iteration in Eq. (6), up to terms of order $O(\mu^2)$.*

This theorem proves that in the limit of infinitesimal step size, a necessary condition for convergence of the algorithm is met. This is not a sufficient condition, but simulations below seem to indicate that convergence actually happens. Note that our analysis assumes that the score function of the noisy data is estimated exactly. To be fully rigorous, some regularity assumptions would also need to be made on the score function but we do not go into such detail.

It is well-known that an infinitesimal step size is necessary in the Langevin algorithm for proper convergence, and a non-infinitesimal step size creates a bias, see e.g., Vempala and Wibisono (2019); this convergence property carries over to our algorithm. In addition, an infinitesimal step size implies that the noise level is infinitesimal, since the two are related by the condition (8).

### 3.2 Special case: Sampling by half-denoising

A particularly interesting special case of the iteration in (6) is obtained when we set the step size

$$\mu = \frac{\sigma^2}{2} \tag{9}$$

This is the smallest $\mu$ allowed according to (8), given a noise level $\sigma^2$. In many cases, it may be useful to use this lower bound since a larger step size in Langevin methods leads to more bias. In fact, Theorem 2 shows, in line with most existing theory, convergence in the limit of infinitesimal step size only.

In this case the noise $\boldsymbol{\nu}_t$ is cancelled, and we are left with a simple iteration:

$$\mathbf{x}_{t+1} = \tilde{\mathbf{x}}_t + \frac{\sigma^2}{2}\mathbf{\Psi}_{\tilde{\mathbf{x}}}(\tilde{\mathbf{x}}_t) \tag{10}$$

with

$$\tilde{\mathbf{x}}_t = \mathbf{x}_t + \sigma\mathbf{n}_t, \quad \mathbf{n}_t \sim \mathcal{N}(\mathbf{0}, \mathbf{I}) \tag{11}$$

where, as above, $\mathbf{\Psi}_{\tilde{\mathbf{x}}}$ is the score function of the noisy data and $\sigma^2$ is the variance of the noise in $\mathbf{\Psi}_{\tilde{\mathbf{x}}}$, typically coming from DSM.

Now, the Tweedie-Miyasawa theorem in Eq. (3) tells us that the optimal nonlinearity to reduce noise is quite similar to (10), but crucially, such optimal denoising has $\sigma^2$ as the coefficient of the score function (for isotropic noise) instead of $\sigma^2/2$ that we have here. Thus, the iteration in Eq. (10) can be called *half-denoising*.

## 4 Experiments

**Data models** We use two different scenarios. The first is *Gaussian mixture model in two dimensions*. Two dimensions was chosen so that we can easily analyze the results by kernel density estimation, as well as plot them. A GMM is a versatile family which also allows for exact sampling as a baseline. The number of kernels (components) is varied from 1 to 4; thus the simulations also include sampling from a 2D Gaussian distribution. The kernels are isotropic, and slightly overlapping. The variances of the variables in the

resulting data were in the range [0.5,1.5]. We consider two different noise levels in the score function, a higher one ($\sigma^2 = 0.3$) and a lower one ($\sigma^2 = 0.1$). These are considered typical in DSM estimation.

The second scenario is *Gaussian model in higher dimensions.* The Gaussian data was white, and the dimension took the values 5,10,100. Only the higher noise level ($\sigma^2 = 0.3$) was considered.

**MCMC algorithms** The starting point is that we only know the noisy score function (i.e. the score function for noisy data), and for one single noise level. Using that, we sample data by the following two methods:

- "Proposed" refers to the sampling using half-denoising and the noisy score function as in (10-11). Assuming the noisy score function is estimated by DSM, this is the method we would propose to generate new data.

- "Basic Langevin" uses the ordinary Langevin method in (5), together with the noisy score function, since that is assumed to be the only one available. This is the (biased) baseline method that could be used for sampling based on DSM and the ordinary Langevin algorithm.

As further baselines and bases for comparison, we use:

- "Oracle Langevin" uses the ordinary Langevin method together with the true score function $\mathbf{\Psi_x}$ (i.e. of the noise-free data $\mathbf{x}$). This is against the main setting of our paper, and arguably unrealistic in deep learning where DSM is typically used. However, it is useful for analyzing the sources of errors in our method.

- "Ground truth" means exact sampling from the true distribution. This is computed to analyze the errors stemming from the methods used in comparing samples, described below.

*Step sizes* are chosen as follows. According to the theory above, the step size for the proposed method is implicitly given based on the noise level as in (9). For the baseline Langevin methods, we use that same step size if nothing is mentioned, but conduct additional simulations with another step size which is 1/4 of the above distribution. This is to control for the fact that the basic Langevin method has the possible advantage that the step size can be freely chosen, and in particular it can be made smaller to reduce the bias due to the non-infinitesimal step size. Note that Oracle Langevin is not identical for the two noise levels precisely because its step size is here defined to be a function of the noise level. Each algorithm was run for 1,000,000 steps.

**Analysis of the sampling results** In the main results (bias removal), the last 30% of the data was analyzed as samples of the distribution; it is assumed here that this 30% represents the final distribution and all the algorithms have converged. (The mixing analysis described next corroborates this assumption.)

In the GMM case, we compute a kernel density estimator for each sampled data set with kernel width 0.1, evaluated at a 2D grid with width of 0.1. We then compute the Euclidean ($L^2$) distance between the estimated densities. In particular, we report the distances of the MCMC method from the ground truth sample, normalized by the norm of the density of the ground truth sample, so that the errors can be interpreted as percentages (e.g. error of -1.0 in $\log_{10}$ scale means 10% error). We further evaluate the accuracy of the distance evaluation just described by computing the distance between two independent ground truth sample data sets (simply denoted by "ground truth" in the distance plots). In the high-dimensional Gaussian case, in contrast, we compute the covariance matrices and use the Euclidean distances between them as the distance measure.

Additional analyses were conducted regarding the mixing speed. We took the GMM with two kernels as above, but ran it in 10,000 trials with 100 steps in each. The algorithm was initialized as a Gaussian distribution with small variance approximately half-way between the two kernels. Here, the convergence was analyzed by looking at the Euclidean distance between the covariance of the sample and the true covariance, for each time point.

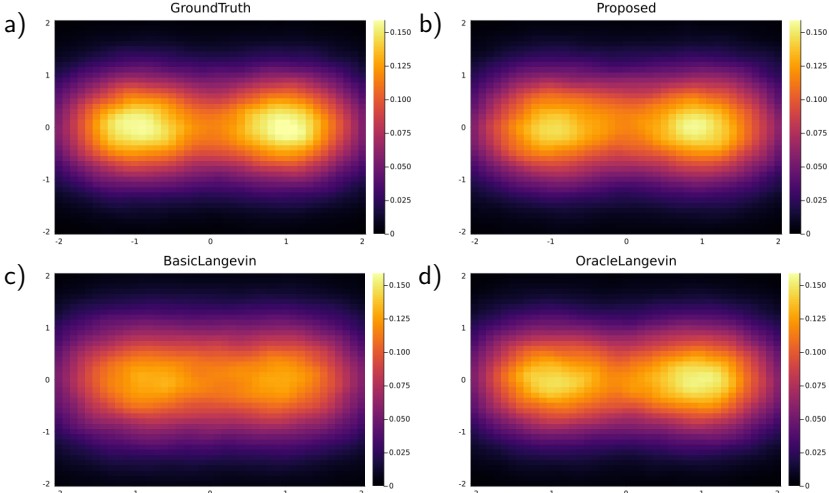

Figure 1: Kernel density estimates for some of the methods, for two kernels in GMM, and lower noise $\sigma^2 = 0.1$ in the score function. The proposed method achieves similar performance to the Oracle Langevin which uses the true score function, thus effectively cancelling the bias due to the noisy score function. The bias due to the non-infinitesimal step size is hardly visible.

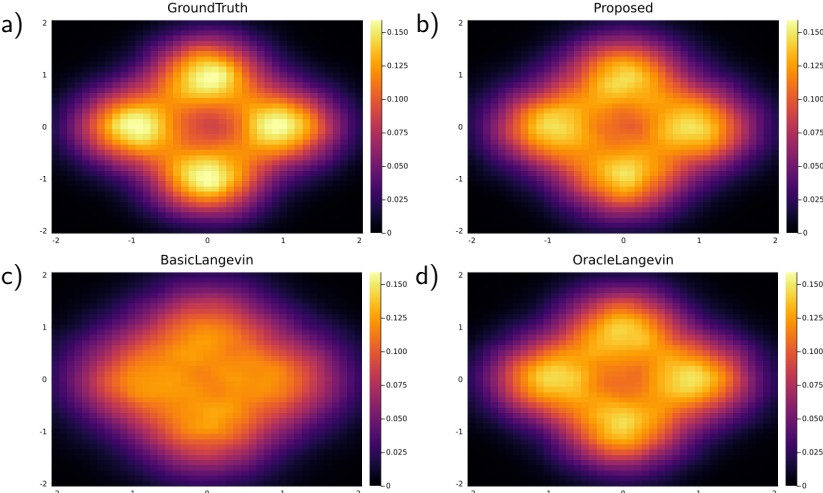

Figure 2: Like Fig. 1 above, but for four kernels in GMM, and a higher noise level $\sigma^2 = 0.3$. Now, some bias due to non-infinitesimal step size is clearly visible even in Oracle Langevin. Still, the proposed method has very similar performance to Oracle Langevin, which is the main claim we wish to demonstrate.

**Results 1: GMM bias removal** Basic visualization of the results is given in Figs 1 and 2. A comprehensive quantitative comparison is given in Fig. 3, which shows that the proposed method clearly outperforms the basic Langevin, thus validating the theoretical result that the effect of noise is reduced.

In fact, the error in the proposed method is very similar to the Oracle Langevin in the case where the step sizes are equal. Most importantly, this shows that almost all the bias in the method is due to the non-infinitesimal step size, as Oracle Langevin suffers from that same bias and has similar performance. Thus, removing the bias due to the noisy score is successful since any possibly remaining noise-induced bias seems to be insignificant compared to the bias induced by the step size. This corroborates the theory of this paper, in particular Theorem 2.

The basic Langevin algorithm has, in principle, the advantage that its step size can be freely chosen; however, we see that this does not give a major advantage (see curves with "mu/4") over the proposed method. In the noisy score case, reducing the step size improves the results of basic Langevin only very slightly; as already pointed out above, the bias due to the noisy score seems to be much stronger than the bias due to non-infinitesimal step size. In contrast, in the case of the Oracle Langevin, we do see that a small step size can sometimes improve results considerably, as seen especially in a), because it reduces the only bias that the method suffers from, i.e. bias due to non-infinitesimal step size. However, a very small step size may sometimes lead to worse results, as seen in b), presumably due to slow convergence or bad mixing. In any case, even Oracle Langevin fails to approach the level of the errors of "ground truth" (i.e. exact sampling, which has non-zero error due to the kernel density estimation).

**Results 2: High-dimensional Gaussian bias removal** The results, shown in Fig. 4, are very similar to the GMM case above. Again, we see that the proposed method has performance which is almost identical to the Oracle Langevin with the same step size. It clearly beats the basic Langevin. Only the Oracle Langevin with a smaller step size is better, and only in high dimensions (arguably, it could be better in low dimensions as well if the step size were carefully tuned and/or more steps were taken).

**Results 3: Mixing speed analysis** Fig. 5 shows that the noise-corrected version mixes with equal speed to the basic Langevin. In fact, the curves are largely overlapping (e.g. solid red vs. solid blue) for the noise-corrected version and the basic Langevin, in the beginning of the algorithm. However, soon the plain Langevin plateaus due to the bias, while the proposed algorithm reduces the error further. Thus, no mixing speed seems to be lost due to the noise-correction.

## 5 Related methods and future directions

It is often assumed that an annealed version of the Langevin algorithm is preferable in practice to avoid the algorithm get stuck in local modes (Lee et al., 2018). A method for annealing the basic Langevin method combined with score matching using different noise levels was proposed by Song and Ermon (2019) and its converge was analyzed by Guo et al. (2024). Our method could easily be combined with such annealing schedules. If the noisy score function were estimated for different noise levels, an annealed method would be readily obtained by starting the iterations using high noise levels and gradually moving to smaller ones. However, a special annealing method could be obtained by using a single estimate of the noisy score, while increasing the parameter $\sigma^2$ in the iteration, now divorced from the actual noise level in $\mathbf{\Psi}_{\bar{\mathbf{x}}}$, from zero to $2\mu$ in the iteration in Eq. (6). We leave the utility of such annealing schedules for future research.

Saremi and Hyvärinen (2019) proposed a heuristic algorithm, called "walk-jump" sampling, inspired by the Tweedie-Miyasawa formula. Basically, they sample noisy data using ordinary Langevin, and then to obtain a sample of noise-free data, they do the full Tweedie-Miyasawa denoising. While there is no proof that this will produce unbiased samples, in further work, Frey et al. (2024); Saremi et al. (2023); Pinheiro et al. (2023) have obtained very good results in real-life application with such an iteration, mostly using an underdamped variant. Related methods have been proposed by Bengio et al. (2013) and Jain and Poole (2022). Likewise, diffusion models use denoising as an integral part. Connections of those methods with our method are an interesting question for future work. However, our method distinguishes itself by denoising only "half-way".

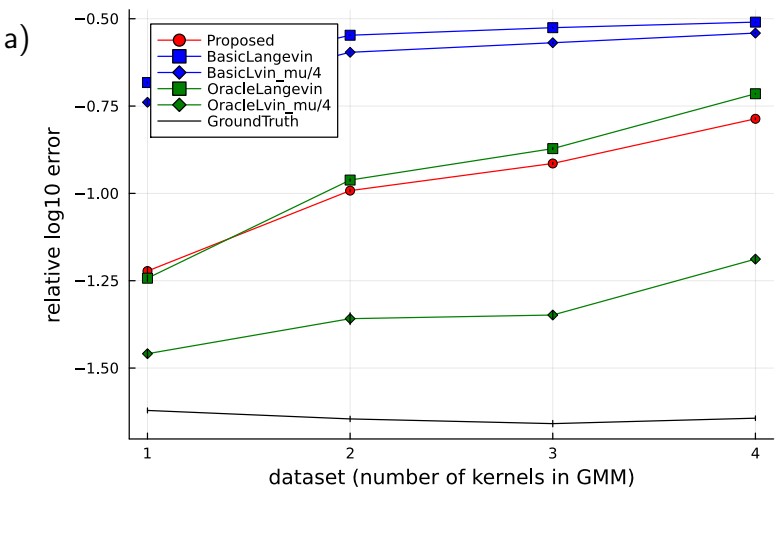

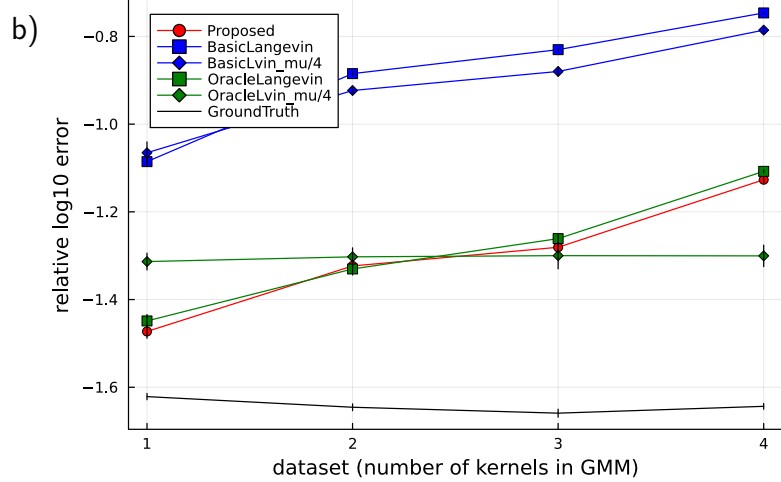

Figure 3: Gaussian 2D mixture model data: Euclidean log-distances between the kernel density estimates for the different sampling methods and the kernel density estimate of the ground truth. a) shows the results for a higher noise level ($\sigma^2 = 0.3$) and b) for a lower noise level ($\sigma^2 = 0.1$) in the score function. Error bars (often too small to be visible) give standard errors of the mean. The curve with label "Ground truth" refers to the distance between two different data sets given by exact sampling; it is far from zero due to errors in the kernel density estimation used in the distance calculation. The different plots for Langevin use different step sizes: "Basic/OracleLangevin" use the same step size as the proposed half-denoising, while the variants with "mu/4" use a step size which is one quarter of that ("Lvin" is a short for "Langevin"). Note that the step sizes for the Oracle Langevin curves are different in a) and b) although the methods are otherwise identical in the two plots; one the other hand, the ground truth is independent of the noise level and thus identical in a) and b).

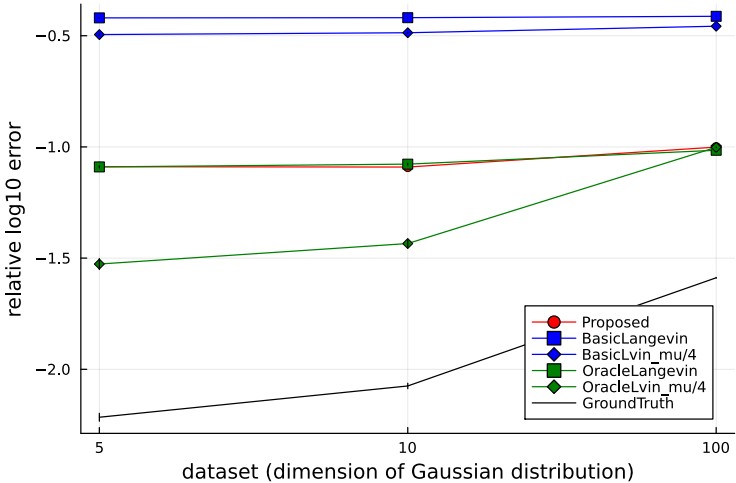

Figure 4: High-dimensional Gaussian data. Euclidean log-distances between the covariance estimates for the different sampling methods and the covariance estimate of the ground truth. Methods and legend as in Fig. 3, here shown for higher noise level ($\sigma^2 = 0.3$) only. ("Proposed" curve may be difficult to see since it is largely under the OracleLangevin curve.)

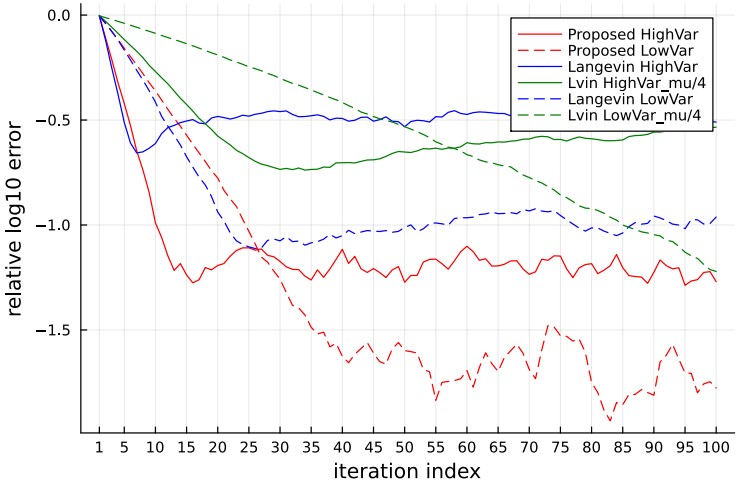

Figure 5: Analysis of mixing speed. The Euclidean log-distances (errors) as a function of iteration count, averaged over 10,000 runs, for a GMM with two kernels. Algorithm legends as in Fig. 3, but here we have the two different variance levels ("HighVar" and "LowVar"') in a single plot.

Some further connection might be found between our method and underdamped Langevin methods (Dockhorn et al., 2021). In particular, an underdamped version of our method would be an interesting direction for future research.

## 6 Conclusion

In generative deep learning, the score function is usually estimated by DSM, which leads to a well-known bias. If the bias is not removed, any subsequent Langevin algorithm (or any ordinary MCMC) will result in biased sampling. In particular, the samples will come from the noisy distribution, that is, the original data with Gaussian noise added, which is obviously unsatisfactory in most practical applications. Therefore, we show here how to remove that bias, i.e., how to sample from the original data distribution when only the biased ("noisy") score function learned by DSM is available. In contrast to, for example, diffusion models, only the score function for one noise level is needed. We prove the convergence in the usual limit cases (infinite data in learning the score function, infinitesimal step size in the sampling). Simple experiments on simulated data confirm the theory.

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

## A    Proof of Theorem 1 (Tweedie-Miyasawa)

We denote the pdf of $\mathbf{n}$ for notational simplicity as

$$\phi(\mathbf{n}) = \frac{1}{(2\pi)^{d/2}|\mathbf{\Sigma_n}|^{1/2}} \exp(-\frac{1}{2}\mathbf{n}^T\mathbf{\Sigma_n}^{-1}\mathbf{n}) \tag{12}$$

To prove the theorem, we start by the simple identity

$$p_{\tilde{\mathbf{x}}}(\tilde{\mathbf{x}}) = \int p(\tilde{\mathbf{x}}|\mathbf{x})p_{\mathbf{x}}(\mathbf{x})d\mathbf{x} = \int \phi(\tilde{\mathbf{x}} - \mathbf{x})p_{\mathbf{x}}(\mathbf{x})d\mathbf{x} \tag{13}$$

Now, we take derivatives of both sides with respect to $\tilde{\mathbf{x}}$ to obtain

$$\nabla_{\mathbf{x}}p_{\tilde{\mathbf{x}}}(\tilde{\mathbf{x}}) = \int \mathbf{\Sigma_n}^{-1}(\mathbf{x} - \tilde{\mathbf{x}})\phi(\tilde{\mathbf{x}} - \mathbf{x})p_{\mathbf{x}}(\mathbf{x})d\mathbf{x} \tag{14}$$

We divide both sides by $p_{\tilde{\mathbf{x}}}(\tilde{\mathbf{x}})$ and rearrange to obtain

$$\mathbf{\Psi}_{\tilde{\mathbf{x}}}(\tilde{\mathbf{x}}) = \int \mathbf{\Sigma_n}^{-1}\mathbf{x}\phi(\tilde{\mathbf{x}} - \mathbf{x})p_{\mathbf{x}}(\mathbf{x})/p_{\tilde{\mathbf{x}}}(\tilde{\mathbf{x}})d\mathbf{x} - \int \mathbf{\Sigma_n}^{-1}\tilde{\mathbf{x}}\phi(\tilde{\mathbf{x}} - \mathbf{x})p_{\mathbf{x}}(\mathbf{x})/p_{\tilde{\mathbf{x}}}(\tilde{\mathbf{x}})d\mathbf{x}$$
$$= \mathbf{\Sigma_n}^{-1}\int \mathbf{x}p(\mathbf{x}|\tilde{\mathbf{x}})d\mathbf{x} - \mathbf{\Sigma_n}^{-1}\tilde{\mathbf{x}}\int p(\mathbf{x}|\tilde{\mathbf{x}})d\mathbf{x} \tag{15}$$

where the last integral is equal to unity. Multiplying both sides by $\mathbf{\Sigma_n}$, and moving the last term to the left-hand-side, we get (3). □

## B Proof of Theorem 2

Define a random variable as

$$\mathbf{y}_t = \tilde{\mathbf{x}}_t + \mu \mathbf{\Psi}_{\tilde{\mathbf{x}}}(\tilde{\mathbf{x}}_t) \tag{16}$$

Assume we are at a point where $\mathbf{x}_t \sim \mathbf{x}$ and thus also $\tilde{\mathbf{x}}_t \sim \tilde{\mathbf{x}}$. We can drop the index $t$ for simplicity. This confounds the $\mathbf{x}_t$ in the algorithm and $\mathbf{x}$ as in the original data being modelled (and likewise for the noisy case), but the distributions are the same by this assumption so this is not a problem.

Consider the characteristic function (Fourier transform of pdf) of $\mathbf{y}$, and make a first-order approximation as:

$$\hat{p}_{\mathbf{y}}(\boldsymbol{\xi}) = \mathrm{E}_{\mathbf{y}} \exp(i\boldsymbol{\xi}^T \mathbf{y}) = \mathrm{E}_{\tilde{\mathbf{x}}} \exp(i\boldsymbol{\xi}^T \tilde{\mathbf{x}}) \exp(\mu i\boldsymbol{\xi}^T \mathbf{\Psi}_{\tilde{\mathbf{x}}}(\tilde{\mathbf{x}})) = \mathrm{E}_{\tilde{\mathbf{x}}} \exp(i\boldsymbol{\xi}^T \tilde{\mathbf{x}})[1 + \mu i\boldsymbol{\xi}^T \mathbf{\Psi}_{\tilde{\mathbf{x}}}(\tilde{\mathbf{x}})] + O(\mu^2) \tag{17}$$

where the last equality holds under some regularity conditions, which we will not try to explicate here. Elementary manipulations further give:

$$\mathrm{E}_{\tilde{\mathbf{x}}} \exp(i\boldsymbol{\xi}^T \tilde{\mathbf{x}})[1 + \mu i\boldsymbol{\xi}^T \mathbf{\Psi}_{\tilde{\mathbf{x}}}(\tilde{\mathbf{x}})] = \hat{p}_{\tilde{\mathbf{x}}}(\boldsymbol{\xi}) + \mu i \int \exp(i\boldsymbol{\xi}^T \tilde{\mathbf{x}}) \sum_{j=1}^{k} \xi_j \frac{\partial p_{\tilde{\mathbf{x}}}(\tilde{\mathbf{x}})}{\partial \tilde{x}_j} d\tilde{\mathbf{x}} \tag{18}$$

Now, we use the well-known trick of integration by parts, and we obtain

$$\sum_{j=1}^{k} \int \exp(i\boldsymbol{\xi}^T \tilde{\mathbf{x}}) \xi_j \frac{\partial p_{\tilde{\mathbf{x}}}(\tilde{\mathbf{x}})}{\partial \tilde{x}_j} d\tilde{\mathbf{x}} = -\sum_{j=1}^{k} i\xi_j^2 \int \exp(i\boldsymbol{\xi}^T \tilde{\mathbf{x}}) p_{\tilde{\mathbf{x}}}(\tilde{\mathbf{x}}) d\tilde{\mathbf{x}} = -i\|\boldsymbol{\xi}\|^2 \hat{p}_{\tilde{\mathbf{x}}}(\boldsymbol{\xi}) \tag{19}$$

And thus,

$$\hat{p}_{\mathbf{y}}(\boldsymbol{\xi}) = \hat{p}_{\tilde{\mathbf{x}}}(\boldsymbol{\xi}) - \mu i^2 \|\boldsymbol{\xi}\|^2 \hat{p}_{\tilde{\mathbf{x}}}(\boldsymbol{\xi}) + O(\mu^2) = \hat{p}_{\tilde{\mathbf{x}}}(\boldsymbol{\xi}) \exp(\mu\|\boldsymbol{\xi}\|^2) + O(\mu^2) \tag{20}$$

We have, using the well-known formula for the characteristic function of Gaussian isotropic noise of variance $\sigma^2$:

$$\hat{p}_{\tilde{\mathbf{x}}}(\boldsymbol{\xi}) = \hat{p}_{\mathbf{x}}(\boldsymbol{\xi}) \exp(-\frac{1}{2}\sigma^2\|\boldsymbol{\xi}\|^2) \tag{21}$$

Thus, we can calculate the characteristic function of $\mathbf{x}_{t+1}$, which we denote by $\hat{p}_+$, by multiplying $\hat{p}_{\mathbf{y}}$ by the characteristic function of the Gaussian isotropic noise with variance $2\mu - \sigma^2$, and obtain

$$\hat{p}_+(\boldsymbol{\xi}) = \hat{p}_{\mathbf{x}}(\boldsymbol{\xi}) \exp(-\frac{1}{2}\sigma^2\|\boldsymbol{\xi}\|^2) \exp(\mu\|\boldsymbol{\xi}\|^2) \exp(-\frac{1}{2}(2\mu - \sigma^2)\|\boldsymbol{\xi}\|^2) + O(\mu^2) \tag{22}$$

Clearly, the terms in the exponents cancel, and thus $\hat{p}_+ = \hat{p}_{\mathbf{x}}$ up to terms of higher order. Thus, the point where $\mathbf{x}_t \sim \mathbf{x}$ is stationary, up to terms of $O(\mu^2)$. $\square$

