# OpenReview forum: "A noise-corrected Langevin algorithm and sampling by half-denoising"
_TMLR — Rejected by TMLR_

### Review · Reviewer_jqkk · 2024-11-04

**Summary Of Contributions:**

The paper proposed a denoising method to handle the noise introduced by the noisy score-matching process. The paper also provides a theoretical analysis of convergence and empirical validation on synthesis datasets.

**Audience:**

Yes

**Claims And Evidence:**

Yes

**Requested Changes:**

+ Could add more references on the topic of Denoising Scoring Matching or Denoising MCMC. As far as I know, there are other denoising MCMC methods pretty similar to the proposed method but just not implemented in the scoring matching.
For example:
Zhang, Ruqi, Andrew Gordon Wilson, and Christopher De Sa. "Low-precision stochastic gradient Langevin dynamics." International Conference on Machine Learning. PMLR, 2022.
The idea is exactly the same, the only difference is that the source of noise is different.
+ Theorem 1 could be a bit challenging to follow, as it requires readers to refer back to the previous context. It would be great to make it a little more self-contained so readers can easily understand it without needing to jump around the paper.
+ The notation $\Psi_{\bf x}(\bf X)$ itself is confusing, the lower case $_\bf X$ should be a random variable. $\bf X$ should be some fixed number and should use another notation.
+ Corollary 1 is also hard to understand. First of $\Psi_{\tilde{x}}$ and $\Psi({\tilde{x}})$ are defined welly. I assume $\Psi_{\tilde{x}}( \cdot ) = \nabla \log p_{\tilde x} (\cdot) $. There is no definition of $\Psi(\cdot).$ So far, I don't think it can be called a corollary.
+ It would be better if the author could provide a proof of corollary 1.
+ Could add experiments on real-world datasets. Since the main application of score-matching methods is image generation problem. Experiments on the synthesis dataset cannot address the concern of noisy scores in that kind of problem.
+ Could provide more precise theoretical analysis. Existing works already provided a non-asymptotic convergence analysis of SGLD on both convex and non-convex cases. The theoretical result provided by this paper is outdated.

**Strengths And Weaknesses:**

## Strengths
Provide convergence analysis.
## Weakness
+ Contributions might be not enough. The denoising idea has been implemented before.
+ Theorems and Corollarys are poorly presented.
+ The theoretical analysis is outdated.
+ Experiment validation is weak and only focuses on synthesis data, the noise is self-defined. Thus it is not that convincing it can handle the real-world noise.

---

> ### Author Response · Authors · 2024-11-21
> **Reply to Review (requested changes)**
>
> Thank you for your many suggestions for improving the presentation, which we can probably all address in a possible revision, including a proof of Corollary 1.
>
> Regarding real data experiments, please see the general reply.
>
> Here are replies to specific questions or requests:
>
> "The denoising idea has been implemented before... The theoretical analysis is outdated." --- We wish to clarify why the contribution of the paper is new. Most existing proofs of Langevin diffusion assume the score function is known. Here, only a *biased* version is known. The work by Zhang, Wilson, De Sa al cited by the reviewer assumes that the score function has noise *added* in it, as in noisy (finite-sample) estimation of the score function, which is very different from a bias (which is due to noise added to the *data*). An approach related to Zhang et al was taken in "A Complete Recipe for Stochastic Gradient MCMC. Yi-An Ma, Tianqi Chen, Emily B. Fox. https://arxiv.org/abs/1506.04696" which was brought to our attention after submission.  In particular, our approach is fundamentally different since estimation of the score function is biased due to using denoising score matching. Thus, we are not aware of any paper that would consider the same problem as our paper.
>
> " Existing works already provided a non-asymptotic convergence analysis of SGLD " --- A non-asymptotic analysis might be useful, but bias is essentially an asymptotic property, it is not clear what the utility of a non-asymptotic analysis would be, and we feel such an analysis is outside of the scope of this paper.
>
> "Corollary 1 is also hard to understand. ... It would be better if the author could provide a proof of corollary 1." --- We can certainly reformulate the Corollary, and provide a proof in the Appendix.
>
> " it is not that convincing it can handle the real-world noise." --- There is no attempt nor claim to handle real-world noise in the paper. We only consider the case where the noise is added for the purpose of self-supervised learning in the particular setting of DSM. Thus, the noise is necessarily artificial (isotropic, Gaussian) in this context.

---

### Review · Reviewer_rYQi · 2024-11-11

**Summary Of Contributions:**

This paper proposes a modified version of the Unadjusted Langevin Algorithm. The goal is to sample from the noise-free distribution with only access to the noisy score function. The authors empirically verify the correctness of the algorithm on synthetic Gaussian mixture problems.

**Audience:**

Yes

**Claims And Evidence:**

No

**Requested Changes:**

1. Enhance theoretical analysis:
	1. Providing convergence analysis would significantly strengthen the paper. If a general proof is challenging, consider offering convergence results for specific cases, such as when the data distribution is Gaussian.
	2. Address the issue of the existence of a stationary distribution before discussing the bias limit in Theorem 2.
	3. Clearly state all regularity assumptions required for the theoretical results to hold, improving the transparency and reproducibility of the work.
2. Add experiments on high-dimensional and non-Gaussian dataset such as non-isotropic Gaussians, MNIST and CIFAR10, to demonstrate the algorithm’s effectiveness in more realistic and challenging scenarios.
3. Relaxing noise assumptions. Discuss how the algorithm could be adapted to handle more general noise models beyond isotropic Gaussian noise.

**Strengths And Weaknesses:**

**Strengths**
1. The paper is well-written and accessible.
2. Sampling from a noise-free distribution given a noisy score function is an interesting and valuable problem with potential applications in various domains.

**Weaknesses**
1. Theoretical justification and Clarity of Theorem 2.
	1. Insufficient guarantee: the proof in Appendix B shows that if the proposed algorithm starts at the noise-free distribution $p(x)$, one iteration of the algorithm stays close to $p(x)$ up to an error $O(\mu^2)$ in the characteristic function. However, a small difference in characteristic function does not guarantee an equally small difference in distribution metrics (like total variance or KL divergence).
	2. Unclear iterative behavior: on the other hand, the proof does not guarantee that arbitrary iterations of the algorithm will remain close to $p(x)$ up to the same bound, raising concerns about the algorithm’s long-term behavior.
	3. The regularity assumptions for Eq. (17) should be made explicit.
	4. It is unclear whether the proposed algorithm possesses a stationary distribution. Without this, claims about convergence to or closeness of the stationary distribution to the target distribution may be incorrect.
2. The paper does not provide theoretical results establishing the convergence of the algorithm, which is crucial for validating its effectiveness and reliability.
3. The assumption that the noise is isotropic Gaussian with known mean and variance is restrictive, limiting its practical use.
4. The experiments are too simple. Both the data and noise distribution are low-dimensional isotropic Gaussians or a mixture of isotropic Gaussian, which may not adequately demonstrate the algorithm’s performance in more complex or high-dimensional settings.

---

> ### Author Response · Authors · 2024-11-21
> **Reply to Review (requested changes)**
>
> Thank you for your time and comments. Here is our reply to the Requested Changes:
>
> " Providing convergence analysis would significantly strengthen the paper. If a general proof is challenging, consider offering convergence results for specific cases, such as when the data distribution is Gaussian." --- The Gaussian case seems feasible, we can probably do it; thank you for the suggestion. Other than that, stronger convergence results would be outside the scope of this paper.
>
> "Address the issue of the existence of a stationary distribution before discussing the bias limit in Theorem 2."  --- We don't quite understand this; how is this different from the point (1.1) above?
>
> " Clearly state all regularity assumptions required for the theoretical results to hold, improving the transparency and reproducibility of the work." --- We believe it is possible to make the theorem rigorous by assuming something like boundedness of the score function. Failing that, the theorem could be downgraded to an intuitive justification of the algorithm.
>
> " Add experiments on high-dimensional and non-Gaussian dataset such as non-isotropic Gaussians, MNIST and CIFAR10, to demonstrate the algorithm’s effectiveness in more realistic and challenging scenarios."  --- The point of real-data experiments is considered in the general reply.
>
> "Relaxing noise assumptions. Discuss how the algorithm could be adapted to handle more general noise models beyond isotropic Gaussian noise." -- In our understanding DSM is always used with Gaussian noise, and probably almost always with isotropic noise. Since the noise is assumed to come from DSM, it does not seem useful to consider more general noise.

---

### Review · Reviewer_V3Ns · 2024-11-11

**Summary Of Contributions:**

This paper presents a new algorithm for sampling from a distribution when having access to the score of noisy data, specifically, the score of the target distribution convolved with a Gaussian measure. The algorithm is a variant of the Langevin algorithm, where instead of the target score, the noisy score is used, and the diffusion coefficient is modified to reduce the bias due to the noisy score. It is proved that the invariant measure of this algorithm is $O(\mu^2)$ closed to the target measure, where $\mu$ is step size. Examples on simple datasets show the benefit of this algorithm in comparison with using basic Langevin with noisy score.

**Audience:**

No

**Broader Impact Concerns:**

Not applicable.

**Claims And Evidence:**

Yes

**Requested Changes:**

* Is there a continuous-time counterpart of Equation (6) that has the original density $p_x$ without any bias as its invariant measure?

* Why do you use a kernel density estimator in Section 4 when we have a closed form for the GMM density?

**Strengths And Weaknesses:**

# Strengths:
The idea of modifying the Langevin algorithm to work with noisy score functions is interesting, especially if it can be used to reduce the computational complexity of modern generative algorithms such as diffusion models which need to learn the score at various noise levels. Furthermore, this paper is accessible to researchers both on the theory and applied side, as it does not use heavy theoretical machinery, or depend on detailed engineering efforts to obtain experimental results.

# Weaknesses:
Overall, I believe the manuscript is not ready for publication. Specifically, I have the following concerns:
* To me, it is not clear where the results of this paper stand in the field. If the goal is to present theoretical results, then there should be theorems that compare the complexity of this algorithm with other methods. There are many theoretical analyses of diffusion models, see e.g. [1], and methods to estimate the score to obtain a concrete sampling algorithm, see e.g. [2]. Is this the only paper that has considered providing theoretical guarantees in a setting where we have access to the score at only one noise level? How do the guarantees compare with diffusion models? What are the theoretical tradeoffs in choosing a single noise level or many noise levels?

* A main success for diffusion models is the fact that they can achieve relatively fast convergence without strong assumptions, e.g. isoperimetry, on the target measure. What is the convergence rate of the algorithm proposed here, e.g. in KL divergence? What are the assumptions required to achieve a convergence rate?

* The statement of Theorem 2 is vague. What is the metric in which the distance is $O(\mu^2)$? What is the dependence on other problem parameters? What are the basic assumptions needed to prove it? For Theorem 1, it might be better to introduce the assumptions in the theorem statement.

* I believe the basic Langevin algorithm is not the only baseline that this algorithm should be experimentally compared to. How about other methods that use denoising score matching?

* The writing of the manuscript can be significantly strengthened. For example, the authors provide a brief discussion on Metrapolis adjustment after introducing the Langevin algorithm which feels unnecessary, given that no references are cited and no other vairants of Langevin are discussed, and that this adjustment is not considered in this paper. Further, the discussion on related works does not provide enough background in either Langevin type sampling algorithms or the use of denoising score matching.

References:

[1] S. Chen et al. "Sampling is as easy as learning the score: theory for diffusion models with minimal data assumptions". ICLR 2023.

[2] X. Huang et al. "Faster Sampling without Isoperimetry via Diffusion-based Monte Carlo". COLT 2024.

---

> ### Author Response · Authors · 2024-11-21
> **Reply to Review**
>
> Thank you for your time and comments. In the following, please find my reply.
>
> "If the goal is to present theoretical results, then there should be theorems that compare the complexity of this algorithm with other methods. There are many theoretical analyses of diffusion models, see e.g. [1], and methods to estimate the score to obtain a concrete sampling algorithm, see e.g. [2]." -- We would like to point out that such complexity analyses are usually done for existing algorithms. We assume that when proposing a new algorithm that solves a new problem, only basic analysis (as in Theorem 2) should be sufficient. Furthermore, please see our general reply regarding the contribution.
>
> "Is this the only paper that has considered providing theoretical guarantees in a setting where we have access to the score at only one noise level?" --- We believe that is the case; we can emphasize that in a possible revised version.
>
> "How do the guarantees compare with diffusion models? What are the theoretical tradeoffs in choosing a single noise level or many noise levels?" -- The starting point of the paper is that we want to develop a method for a single noise level because it is simpler *computationally*; it also fills a gap in the theory. There is no claim that the resulting method would be better than diffusion models in terms of the actual generation quality, assuming no constraints in computation.
>
> "A main success for diffusion models is the fact that they can achieve relatively fast convergence without strong assumptions, e.g. isoperimetry, on the target measure. What is the convergence rate of the algorithm proposed here, e.g. in KL divergence? What are the assumptions required to achieve a convergence rate?" --- Analysis of the convergence rate sound to me like a completely different project. Here, the claim is that the bias can be removed in a Langevin-like algorithm.
>
> "The statement of Theorem 2 is vague. What is the metric in which the distance is? What is the dependence on other problem parameters? What are the basic assumptions needed to prove it?" -- This is not a real "proof of convergence", since it is only about a stationary point, in which case the question about the distance does not apply.  The lack of rigorous regularity conditions is, we admit, a weakness of the paper, at least in its current version. We think that we have meanwhile found conditions that make the proof rigorous, basically assuming boundedness of the score function should be sufficient. If that fails, the theorem could be downgraded to a heuristic justification of the algorithm.
>
> "I believe the basic Langevin algorithm is not the only baseline that this algorithm should be experimentally compared to. How about other methods that use denoising score matching?": This would probably mean diffusion models, which I considered earlier in this reply. Again, the comparison is not meaningful in the sense that diffusion models need more learning/computation.
>
> "The writing of the manuscript can be significantly strengthened. For example, the authors provide a brief discussion on Metrapolis adjustment after introducing the Langevin algorithm which feels unnecessary, given that no references are cited and no other vairants of Langevin are discussed, and that this adjustment is not considered in this paper. Further, the discussion on related works does not provide enough background in either Langevin type sampling algorithms or the use of denoising score matching." -- Certainly, we will be happy to improve the writing and expand the discussion.
>
> "Is there a continuous-time counterpart of Equation (6)..." -- There does not seem to be any obvious counterpart, since the noise is added on x, instead of being added on the deterministic part of the differential; this is very unconventional in SDE.
>
> "Why do you use a kernel density estimator in Section 4 when we have a closed form for the GMM density?" --- We could directly compare the KDE of the sampled data with the theoretical pdf of the GMM, but that would ignore how much of the distance (error) is due to the KDE itself. Therefore, we chose to do KDE on data sampled from GMM as well. This puts all the distributions on the same footing, and enables analyzing how much of the error is due to the KDE as opposed to the sampling itself. In the plots, there is a curve labelled "GroundTruth" which gives exactly that: a rough estimate of how much of the error is due to the KDE process.

---

### Author Response · Authors · 2024-11-14
**General comments on whether our claims are supported by the evidence**

Thank you for the detailed reviews. At this point, it may be useful to discuss some possible disagreement on what is actually needed for publication at TMLR.

We submitted this paper to TMLR in the understanding that a short paper with precise if modest claims would be acceptable, obviously assuming the claims are correct; we saw this as one of the points for creating TMLR in the first place.

Now, it seems that the reviewers largely want us to add new claims and provide evidence for them in order to increase the significance of the contribution.

The TMLR web page says: "Any gap between claims and evidence should be addressed by the authors. Often, this will lead reviewers to ask the authors to provide more evidence by running more experiments. However, this is not the only way to address such concerns. Another is simply for the authors to adjust (reduce) their claims."

We would be happy to reduce the claims as considered necessary. Our claims are that a) the bias is removed in the precise sense that the algorithm has a stationary point at the noise-free distribution, up to higher-order terms,  b) the algorithm needs less information than diffusion models (only one noisy score), and c) in simple simulations, the removal of the bias seems to work. We don't think it would be useful to proceed by making stronger claims on the convergence of the algorithm, utility on real data, superiority to diffusion models, or applicability to different noise models.

We notice now that the conclusion contains a couple of exaggerated claims but this can be fixed easily. We say "We prove the convergence" but this is exaggerated; it would be more appropriate to say something like "We analyze...". Likewise, we say "we show here how to remove that bias" which could also be changed to "we propose here an algorithm to remove that bias". A similar critique can be applied to Theorem 2. As the paper already says, it suffers from some lack of rigor since the regularity conditions are not given. Moreover, as pointed out by the reviewers, the exact meaning of convergence is not given either. The easiest fix would be to simply discuss these limitations and label the theorem as "informal", "non-rigorous", or similar. Making it rigorous would actually complicate the presentation a lot; nor are we sure if it can be done with a reasonable amount of work.

Of course, we will be happy to explain things better, such as Corollary 1 and other points requested by the reviewers. Also, it is possible that some further exaggerated claims can be found in the paper and need to be fixed.

But we're also happy that the reviewers do not seem to find anything actually wrong or incorrect with the results (with the possible exception of the rigor in Theorem 2; but the level of rigor needed is always to some extent subjective).

If a major strengthening of the results is deemed necessary, we would prefer to withdraw the paper: in that case, we would have to conclude that our understanding of the fundamental policies of TMLR is different from those of the reviewers and/or the AE.

---

> ### Comment · Action_Editor_D8gz · 2024-11-19
>
> Dear Authors,
>
> Thank you for expressing your concerns. Please note that the decision will be made in accordance with the TMLR acceptance criteria. However, the reviewers have raised some questions and concerns that should be addressed. Could you kindly respond to those questions? You may, however, skip comments specifically related to significance.
>
> Best regards,
> AE

---

### Decision · Action_Editor_D8gz · 2025-02-11

**Recommendation:** Reject

**Comment:**

Theorem 2 basically proves that the distribution obtained by one iteration of the method starting at the noise free distribution $p_x$ stays close to $p_x$ up to $O(\mu^2)$ error in the characteristic function. While this indicates that the proposed update does not shift $p_x$ significantly, it does not necessarily implies that the stationary distribution approximates $p_x$ w.r.t. a certain metric. To rigorously support the claim, the paper needs to show (i) there exists the stationary distribution that is invariant under the proposed iteration and (ii) the gap w.r.t. a distribution metric between the stationary distribution and $p_x$ is small.

**Audience:**

This paper could be of interest to parts of the TMLR audience.

**Claims And Evidence:**

This paper proposes a new sampling method that leverages a noisy score function, inspired by Langevin Monte Carlo and denoising score matching. To mitigate the bias introduced by the noisy score, the authors also propose a noise-correcting technique. The primary claim is that the original noise-free distribution $p_x$ approximates the stationary distribution of the proposed dynamics up to $O(\mu^2)$ error, where $\mu$ is the step size. However, the current statement (Theorem 2) is somewhat vague, as it does not clarify how the gap, measured with respect to a certain distribution metric, becomes small.

**Resubmission Of Major Revision:**

The authors may consider submitting a major revision at a later time.